# Pan-Omics in Sheep: Unveiling Genetic Landscapes

**DOI:** 10.3390/ani14020273

**Published:** 2024-01-15

**Authors:** Mengfei Li, Ying Lu, Zhendong Gao, Dan Yue, Jieyun Hong, Jiao Wu, Dongmei Xi, Weidong Deng, Yuqing Chong

**Affiliations:** 1Faculty of Animal Science and Technology, Yunnan Agricultural University, Kunming 650201, China; mfli_2000@163.com (M.L.); yinglu_1998@163.com (Y.L.); zander_gao@163.com (Z.G.); danyue0528@foxmail.com (D.Y.); hongjieyun@163.com (J.H.); 15229238680@163.com (J.W.); xidmynau@163.com (D.X.); dengwd@ynau.edu.cn (W.D.); 2Faculty of Animal Science and Technology, Yuxi Agricultural Vocational and Technical College, Yuxi 653106, China

**Keywords:** multiomics integration, sheep, genetic breeding insights

## Abstract

**Simple Summary:**

Panomics refers to the integration of multiple omics technologies to comprehensively examine and interpret biomolecular data at different levels. This approach facilitates the study of sheep genetics and related functions, providing valuable insights into sheep breeding, disease prevention, and treatment strategies. By analyzing the complex interactions between genes, proteins, and metabolites, we can gain a deeper understanding of sheep biology and ultimately improve productivity and reproduction. Panomics, in particular, allows for the prediction, screening, and pinpointing of animal genetic traits, overcoming the limitations of traditional breeding methods like age constraints and issues with accuracy and reliability. This advancement holds immense importance for sheep breeding and agricultural production.

**Abstract:**

Multi-omics-integrated analysis, known as panomics, represents an advanced methodology that harnesses various high-throughput technologies encompassing genomics, epigenomics, transcriptomics, proteomics, and metabolomics. Sheep, playing a pivotal role in agricultural sectors due to their substantial economic importance, have witnessed remarkable advancements in genetic breeding through the amalgamation of multiomics analyses, particularly with the evolution of high-throughput technologies. This integrative approach has established a robust theoretical foundation, enabling a deeper understanding of sheep genetics and fostering improvements in breeding strategies. The comprehensive insights obtained through this approach shed light on diverse facets of sheep development, including growth, reproduction, disease resistance, and the quality of livestock products. This review primarily focuses on the application of principal omics analysis technologies in sheep, emphasizing correlation studies between multiomics data and specific traits such as meat quality, wool characteristics, and reproductive features. Additionally, this paper anticipates forthcoming trends and potential developments in this field.

## 1. Introduction

Sheep represent one of the most widespread livestock genetic resources globally, with over 1400 distinct breeds found worldwide [1,2]. Their domestication traces back approximately 10,000 years, believed to have originated in the Middle East or Mesopotamia in Asia [3]. Known for their docile nature, sheep offer valuable resources, providing wool for clothing and high-quality meat with significant taste and nutritional value, thus serving as a crucial source of sustenance and textiles for humans [4]. Due to the considerable genetic diversity among sheep, the implementation of comprehensive multiomics approaches in studying their genetic information and associated functionalities across various levels plays a pivotal role in improving their productivity and reproductive performance.

Multiomics joint analysis, also known as panomics and rooted in Crick’s seminal concept of the DNA–RNA–protein axis, synergistically integrates two or more single-omics technologies. This integration facilitates comprehensive comparative analysis, aimed at decoding the intricate regulation of genetic information across diverse biomolecules [5]. It delves into the dynamics of differential expression and unveils interconnections across various biomolecular layers, offering profound insights into biological functions and physiological mechanisms of living organisms. Acknowledging the presence of molecular disparities at multiple levels of gene regulation—encompassing genomic variations, gene expression, protein translation, and post-translational modifications—is critical [6]. These dynamic systemic changes interact intricately, contributing to the complexity of the biological landscape [7]. Furthermore, tailored approaches for omics data analysis are imperative due to genetic differences among biological samples. Multiomics technologies, spanning genomics, epigenomics, transcriptomics, proteomics, metabolomics, microbiomics, and beyond [8,9,10], furnish a comprehensive toolkit for understanding an organism’s biological traits and disease foundations [11]. By integrating insights across various research levels (as depicted in Figure 1), a multiomics approach consolidates data from distinct omics layers, enabling the exploration of their intricate interactions in organismal research [12]. Multiomics correlation analysis aids in predicting, screening, and pinpointing genetic information within animal organisms, surmounting limitations of traditional breeding methods in terms of efficiency, precision, and reliability. This research yields crucial insights for sheep breeding, disease prevention, and treatment strategies. By elucidating the intricate interactions between genes, proteins, and metabolites, a deeper understanding of sheep biology emerges, offering scientific guidance for breed enhancement and increased agricultural production efficiency. This paper aims to summarize recent advancements in utilizing multiomics technologies for sheep studies, intending to guide future research directions and lay the foundation for further exploration.

## 2. Main Omics Analysis Techniques

### 2.1. Genomics

Genomics constitutes an expansive field aimed at unraveling the intricate organization, function, and variability of genetic information within organisms [13]. Its primary objectives encompass understanding the correlation between genes and phenotypes, investigating genome evolution, exploring genetic diversity, and elucidating genome regulation [14]. This involves comprehensive examination and analysis of an organism’s complete genome, offering insights into its structural organization, functionality, and diversity. The burgeoning realm of genomics serves as a pivotal framework for pinpointing specific genetic variations associated with Mendelian genetic disorders and complex diseases [15], such as the Illumina BeadChip, a gene chip technology that simultaneously detects tens of thousands of genes or single-nucleotide polymorphisms (SNPs) and analyzes the expression of genes or SNPs, genetic variants, genotypes, and other information in a sample [16]. Noteworthy methodologies in current genomic research encompass de novo sequencing), resequencing, and streamlined genome sequencing. The advent of high-throughput sequencing technology has notably propelled genomics, facilitating the efficient acquisition and analysis of vast genomic datasets.

The diversity and richness inherent in sheep breeds, characterized by economically valuable traits such as high-quality wool, multiribbed features, distinctive teat characteristics, and remarkable stress resistance, have garnered intense research interest [17]. Studying phenotypic transformations during sheep domestication assumes paramount importance in refining breeding practices. Genome sequencing offers comprehensive analyses of genetic variations, population structures, and trait selections within sheep breeds [18]. In the foreseeable future, a well-defined genome map will play a pivotal role in applying genomic selection techniques for animal breeding purposes [19]. Since the publication of the first reference genome of the Texel sheep breed [20], the meticulous assembly of various sheep breed reference genomes (as detailed in Table 1) has significantly expanded the genomic landscape. These genomes serve as invaluable resources for identifying crucial functional genes associated with growth, development, carcass quality, meat characteristics, reproduction, and disease resistance in sheep. Highlighting the impact of sheep genomics, multiple studies have unveiled substantial genetic insights.

By exploring the complexities within sheep *ASIP* gene expression, an insightful analysis unveiled a 190 kb tandem repeat spanning the sheep *ASIP* and *AHCY* coding regions, alongside the *ITCH* promoter region. This region plays a pivotal role in determining coat color, where a duplicated copy of the *ITCH* promoter regulating the second copy of the *ASIP* coding sequence is associated with the dominant white sheep phenotype. Conversely, recessive black sheep exhibit a single-copy *ASIP* gene with a silenced *ASIP* promoter, revealing nonallelic homologous recombination and gene mutation at the *ASIP* locus as key factors in the evolution of sheep pigmentation [24]. Additionally, in a study of the 3′ end of sheep *RXFP2*, a 4 kb region was amplified, revealing a 1833 bp insertion in the 3′ untranslated region (3′UTR) of *RXFP2* in hornless sheep [25]. The study proposed a PCR-based genotyping method to determine hornless genotypes in sheep, shedding light on the role of sequence insertions in altering gene expression. In the pursuit of unraveling the mysteries surrounding rib numbers in Hu sheep, a genome-wide association study (GWAS) highlighted 219 single-nucleotide polymorphism loci overlapping with 206 genes. These genes are primarily associated with rib development processes, inorganic anion transport, cellular biosynthesis processes, the oxytocin signaling pathway, and the regulation of arrhythmogenic right ventricular cardiomyopathy. Notably, *CPOX* (fecal porphyrinogen oxidase), *KCNH1* (potassium voltage-gated channel, subfamily H, member 1), and *CPQ* (carboxypeptidase Q) genes were identified as jointly influencing rib number in Hu sheep [26]. In a recent groundbreaking effort, the pan-genome of sheep was meticulously constructed, leading to the profound discovery of a mutation in the 5′ untranslated region (5′UTR) of the *HOXB13* gene. This mutation was found to result in long-tailed traits, unveiled through a comprehensive genome-wide association study and gene expression analysis [27]. Sheep milk is widely utilized in the production of various cultured dairy products. Through whole-genome sequencing analysis, significant genomic regions and genes related to milk production traits have been identified. The most notable genes associated with these traits include *ST3GAL1*, *CSN1S1*, *CSN2*, *OSBPL8*, *SLC35A3*, *VPS13B*, *DPY19L1*, *CCDC152*, *NT5DC1*, *P4HTM*, *CYTH4*, *METRNL*, *U1*, *U6*, and *5S_rRNA* [28]. These findings provide the genetic structure of milk production and composition in sheep with deeper insights. Whole-genome sequencing of sheep was conducted to investigate their adaptation to extreme environments, employing large-effect SNP analysis of candidate genes in Tibetan and Taklamakan desert region breeds. This approach revealed a variety of novel genes, important pathways, and GO categories associated with localized adaptations in sheep in plateau and desert environments [29]. Utilizing whole-genome sequences, single-nucleotide polymorphism arrays, mitochondrial DNA, and Y-chromosome variation, we studied genomic variation in 986 samples of Tibetan sheep across their range. The results shed light on the evolutionary mechanism of adaptive gene infiltration from pan sheep to Tibetan sheep, exemplified by genes such as *HBB*, *HBE*, and *RXFP2*, which explains their rapid local adaptation [30].

Summarily, genomic studies in sheep have identified candidate genes linked to long tails, hornless traits, cashmere yield, wool color, and rib count (Appendix A). These discoveries establish a robust foundation and offer a potent tool for enhancing genetic improvement and increasing sheep production efficiency. A comprehensive exploration of the sheep genome holds promise in uncovering pivotal genes and intricate molecular mechanisms governing economically valuable traits. While genomics-driven investigations excel in pinpointing differentially expressed genes with scientific precision, the inherent diversity among sheep species yields a plethora of genetic variations. However, this rich diversity, while invaluable, might not provide a comprehensive depiction of the overall attributes and intricate interplays within sheep biological systems.

### 2.2. Epigenomics

Epigenomics investigates intricate genetic regulation mechanisms involving chemical modifications and spatial structural changes within the genome. These modifications exert influence over gene functionality and expression, independent of alterations in the nucleotide sequence. Fundamental epigenetic controls encompass histone modifications, DNA methylation, and RNA modifications [31,32]. Histone modification encompasses various enzymatic alterations, including methylation, acetylation, phosphorylation, adenylation, ubiquitination, and ADP-ribosylation, all pivotal in governing gene regulation. DNA methylation, catalyzed by DNA methyltransferases [33,34], entails the addition of methyl groups to DNA’s CpG sites, significantly impacting gene expression. RNA modification, particularly mRNA modification, occurs at the transcriptional level and crucially regulates gene expression [35]. Epigenomic investigations are crucial for comprehending the role of epigenetic regulation in sheep development and health [36].

Using whole-genome bisulfite sequencing (WGBS), researchers scrutinized the DNA methylation profiles of ovarian DNA in Hu sheep, illuminating intricate epigenetic landscapes. The study pinpointed 10 differentially methylated genes (DMGs) associated with fecundity, underscoring the significance of DNA methylation in comprehending sheep reproductive capabilities [37]. Employing WGBS, researchers meticulously examined DNA methylation patterns during muscle growth in Hu sheep, unveiling insights into the dynamic epigenetic modulation of this biological process. The investigation identified nine DMGs linked to muscle development and metabolism, with genes like *MAPT*, *DIAPH1*, *NR4A1*, and *DLK1* emerging as pivotal regulators in skeletal muscle development [38]. The DNA methylation of these genes plays a pivotal role in muscle development, influencing sheep’s muscular growth. Additionally, in a recent study, researchers scrutinized the effects of nutritional supplementation during pregnancy on ewe offspring, shedding light on the repercussions of maternal nutrition on the subsequent generation. The findings revealed that both undernutrition and overnutrition could induce alterations in small noncoding RNA (sncRNA) and DNA methylation in F_1_ offspring’s spermatozoa, leading to epigenetic modifications in the fetus that affect development and productivity [39].

In summary, the utilization of epigenomic tools in sheep research is paving the way for augmenting productivity, product quality, and health in these animals (Appendix A). This deepens our understanding of epigenetic regulatory mechanisms in sheep and offers valuable insights for leveraging epigenetic modifications to advance sheep genetics. However, it is still necessary to overcome the complexity of data analysis, challenges in data interpretation, sample selection, and environmental impacts in order to achieve better research and application results.

### 2.3. Transcriptomics

Transcriptomics, a specialized field within genomics, explores the intricacies of gene transcription within cells and comprehensively studies the governing principles of transcriptional regulation [40]. Transcriptome sequencing, commonly known as RNA-seq, captures the entire spectrum of transcripts within a cell during specific developmental stages or physiological conditions [41]. This technique represents a reservoir of biological information, encompassing gene expression levels, structural features, antisense transcripts, alternative splicing, single nucleotide polymorphisms, and gene fusions. Transcriptomics plays a fundamental role in deciphering the functional components of the genome and understanding the molecular mechanisms driving cellular and tissue function [42]. It is pivotal in studies related to biological phenotypes and the dynamics of gene expression [43]. Through transcriptomic analyses, researchers uncover insights into gene expression modulation under various conditions, pathway activation, cellular function alterations, and gene expression patterns associated with diverse diseases.

Transcriptomic studies in sheep offer invaluable insights into gene expression and its intricate regulation. For instance, RNA-seq analysis scrutinized the regulatory mechanisms of mRNAs and lncRNAs linked to prolificacy-related genes in sheep. Investigating the pituitary gland of high- and low-prolificity sheep unveiled 57 differentially expressed lncRNAs and 298 differentially expressed mRNAs. Further insights into the interaction between the candidate lncRNA MSTRG.259847.2 and its target gene *SMAD2* were substantiated in sheep pituitary cells [44]. By utilizing RNA-seq, an investigation into the unsaturated fatty acid content in sheep muscle uncovered mutations in genes like *APOA17*, *CFHR5*, *TGFBR5*, and *LEPR* associated with fatty acid composition [45]. Moreover, RNA-seq revealed differentially expressed mRNAs in the hypothalamus, pituitary, and ovary of Small-Tailed Han sheep and Tan sheep, exposing key genes implicated in estrogenic processes. The study identified a total of 2569, 2704, and 4156 significantly differentially expressed genes in the hypothalamus, pituitary, and ovary, respectively [46]. Different genotypes of *FecB* in sheep display varying ovulation rates and fecundity due to the secretion of different reproductive hormones by the hypothalamic–pituitary–ovarian axis. Transcriptome sequencing was employed to analyze the expression of the hypothalamus during the follicular and luteal phases in sheep with different genotypes. This analysis identified 53 differentially expressed mRNAs (DEGs) and 40 differentially expressed long-chain noncoding RNAs (DELs). Among these, two DEGs (*FKBP5* and *KITLG*) showed enrichment in melanogenesis, oxytocin, and GnRH secretion pathways. Additionally, LINC-219386 and IGF2-AS exhibited high expression levels in *FecB* mutant sheep and regulated their target genes (*DMXL2* and *IGF2*) to promote increased production of GnRH during follicular follicle development [47]. Lamb meat with lower fat content is currently preferred by consumers. However, the significant energy consumption of tail fat during rearing has a noticeable impact on the profitability of livestock farming businesses. To address this issue, identifying and selecting key genes that influence tail fat deposition, as well as studying the molecular regulatory mechanisms controlling fat deposition, can provide valuable insights for genetic breeding and selection in sheep farming enterprises [48]. Employing RNA-seq, pivotal microRNAs implicated in fat deposition across various tailed sheep breeds were identified. Among the 155 differentially expressed miRNAs discovered, miR-379-5p and *HOXC9* exhibited disparate expression patterns in the tail adipose tissue of Tibetan sheep and Hu sheep [49]. This finding lays a theoretical foundation for studying tail adipogenesis in sheep. CircRNA is a subtype of noncoding RNA that has been demonstrated to play a crucial role in the function of the mammary gland (MG). RNA-Seq was employed to profile circRNA expression in the MG of sheep, uncovering variations in milk yield and composition phenotypes. A total of 4906 circRNAs were detected, with 33 of them exhibiting differential expression between different sheep breeds [50]. These results may contribute to a deeper comprehension of the mechanisms underlying circRNA function in MG development and milk secretion in sheep.

In summary, transcriptomics in sheep research has been pivotal in revealing insights into various traits such as tail fat, meat quality, and reproductive processes (Appendix A). The extensive use of transcriptomics facilitates the discovery of gene regulatory networks, functional genes, and genes associated with various diseases. This provides an essential database and theoretical framework for understanding sheep physiology, development, and adaptability. Nevertheless, there are still challenges such as complex data analysis, RNA modification and different splicing forms, limitations in sample sources, and difficulties in detecting low-expression-level genes.

### 2.4. Proteomics

Proteomics, dedicated to the comprehensive study of proteins, plays a pivotal role in understanding biological functions [51]. This field goes beyond the mere identification and quantification of proteins in cells, tissues, or organisms, offering complementary insights alongside genomics and transcriptomics [52]. Proteins, as active agents in biological functions, are influenced not only by mRNA levels but also by translational control and host regulation [53]. Proteomics offers a detailed understanding of protein expression, structure, function, interactions, and modifications across various stages, crucial for unraveling complex biochemical processes at the molecular level [54,55]. The proteome is dynamic, responding to both intracellular and external stimuli. Through proteomic analysis, researchers discern changes in gene expression, differentiating between distinct biological states of a cell [56,57]. Exploring protein properties and functions enables the comprehension of intricate biological processes in organisms, paving the way for discovering novel therapeutic approaches and drug targets.

In the context of sheep research, proteomics presents diverse applications for examining the composition and function of sheep proteome. For example, in a protein profiling initiative, the dorsal longissimus muscle of Chinese Merino sheep at the embryonic ages of 85, 105, and 135 days was analyzed. A total of 5520 proteins were successfully identified, with 1316 displaying differential abundance. These findings highlighted that the period from 85 to 105 days marks the proliferation of embryonic muscle fibers, while the stage from 105 to 135 days signifies their hypertrophy [58]. To discern differentially expressed proteins (DEPs) associated with tail phenotypes, advanced proteomics technology identified 3248 proteins, among which 44 were upregulated and 40 were downregulated DEPs. Notably, APOA2, GALK1, ADIPOQ, and NDUFS4 were implicated in sheep tail fat deposition and metabolism [59]. A comparative proteomic analysis of rumen epithelial tissues across different sheep ages revealed 4523 proteins, indicating the involvement of processes like glutathione, the Wnt signaling pathway, and the Notch signaling pathway in rumen epithelial cell growth [60]. Moreover, the ubiquitination and post-translational modification of histones emerged as critical molecular elements regulating rumen epithelial development. Employing label-free proteomics, researchers investigated genetic factors influencing body weight in sheep, identifying differentially abundant proteins (DAPs) in Hu sheep and Dorper sheep, linking several DAPs to immune response, fat deposition, and muscle development [61]. The study suggests that body weight regulation in sheep involves multiple pathways, and these DAPs could serve as potential markers for predicting sheep body weight.

Overall, the application of proteomics in sheep research significantly advances our understanding of sheep biology, protein expression dynamics, and disease mechanisms (Appendix A). It offers invaluable scientific support for enhancing animal husbandry practices and furthering human health. But at the same time, it also faces challenges such as difficulties in data interpretation and high technical costs.

### 2.5. Metabolomics

Metabolomics involves the identification, quantification, and analysis of metabolites present in biological fluids, cells, and tissues [62,63]. This field employs techniques such as nuclear magnetic resonance, mass spectrometry, and vibrational spectroscopy [64]. Notably, metabolomics is instrumental in biomarker discovery, owing to its sensitivity in detecting subtle changes within biological pathways [65]. Metabolites, essential for cellular function, encompass metabolic substrates and products crucial for cellular processes [66]. Aimed at providing a comprehensive analysis of low-molecular-weight molecules within organisms [63,67,68], metabolomics offers distinct advantages over other omics methods by focusing on the intermediary products of gene and protein expression [69,70]. This approach unveils insights into an organism’s health status, the regulatory mechanisms of metabolic pathways, and the impacts of environmental factors, laying a substantial scientific groundwork for applications in disease diagnosis, drug development, crop improvement, and food safety.

For instance, a comparative analysis of liver and muscle metabolomes in Merino, Damara, and Dorper sheep revealed responses to feed restriction. The study showcased that Damara and Dorper sheep exhibited enhanced tolerance to seasonal weight loss (SWL), a critical trait for adaptation to challenging environmental conditions [71]. This adaptability to SWL significantly reduces yield, highlighting the economic significance of this finding. Using nontargeted and targeted metabolomics, meat quality and differential metabolites in Tibetan sheep were scrutinized. Nontargeted metabolomes were analyzed using UHPLC-QTOF-MS, while targeted metabolomes focused on amino acids, assessed through UHPLC-QQQ-MS, and fatty acids were analyzed using GC-MS. The results reveal significant correlations between nontargeted metabolomics outcomes and phenotypic data related to meat quality, including shear, cooking loss, drip loss, chewiness, elasticity, flavor, and protein and fat content [72]. Examining the dynamic changes in metabolites and metabolic pathways in Mongolia ovine during postmortem freezing aging identified a total of 1093 metabolites, with 467 displaying significant changes during aging. These alterations encompassed amino acids and their metabolites, fatty acyl groups, and glycerophospholipids. The findings suggested enriched metabolic pathways during aging, including protein digestion and absorption, aminoacyl-tRNA biosynthesis, unsaturated fatty acid biosynthesis, nucleotide metabolism, and carbon metabolism [73]. A comparison of metabolomics between sheep’s and goat’s milk showed that sheep’s milk had higher protein, fat, and lactose content [74]. These findings provide valuable insights into the compositional differences between these two types of milk, which can have implications for dietary considerations and product development in the dairy industry.

In summary, metabolomics offers crucial insights into how sheep adapt to external environments and regulate their metabolism (Appendix A). This understanding significantly advances our knowledge of sheep biology and breeding. However, metabolomics can only reflect the overall state of intracellular metabolism and cannot analyze the interactions between specific molecules, which poses certain limitations on the mechanism of disease occurrence and the determination of therapeutic targets.

The summarized principles of single omics and their applications in sheep genetic breeding, as discussed above, are presented in Table 2.

## 3. Research and Implementation of Multiomics Integration in Sheep Production

### 3.1. Meat Traits

Sheep meat traits, increasingly pivotal in sheep breeding programs, arise from a complex interplay of genetic and environmental factors. Recent strides in omics techniques have revolutionized the analysis of proteins and metabolites in sheep meat, elevating the precision of quality assessment and confirming meat authenticity at a molecular level. These advancements hold significant promise for augmenting sheep meat quality [75]. In similar environmental contexts, genetic background emerges as a critical determinant influencing meat production [76]. Integrated multiomics analysis promises a more comprehensive understanding of meat quality trait development, quality enhancement, and the identification of genes linked to sheep meat quality.

For example, a comprehensive molecular exploration through a comparative RNA-seq and proteomics analysis of pectoral muscle tissues from Hu sheep and Dorper sheep unveiled 22 DEGs and proteins (DEPs) associated with lipid transport, metabolism, and muscular phylogeny [77]. This analysis showcased analogous trends in mRNA and protein expression, offering a holistic perspective on the molecular underpinnings of meat traits in sheep, spanning from gene expression to the protein level. Another study examining perirenal tissues from Assaf sheep employed RNA sequencing and whole-genome bisulfite sequencing (WGBS). This multiomics approach identified 314 genes and 627 differentially methylated regions within these genes, differentiating between males and females. Moreover, it pinpointed differential coexpression (DcoExp) gene modules between genders, featuring 22 selected genes potentially influencing fat and meat quality characteristics due to sex differences [78]. This highlights the potential of RNA sequencing and WGBS technology in exploring sex-related characteristics in sheep.

Fat deposition profoundly influences lamb flavor. The functional role and mechanism of bile acids in lamb fat deposition were scrutinized using RNA-seq and targeted metabolomics. This analysis unveiled differential genes associated with ferritin and fatty acid biosynthesis, including *HSPA8*, *HIF1A*, *HEXB*, *ACSL6*, and *MAP1LC3B*, linked to tail lipid weight and proportions, potentially serving as marker genes for tail lipid regulation [79]. Additionally, an investigation utilizing RNA-seq and metabolome association analysis identified key genes governing muscle flavor precursors in sheep. The findings underscored the importance of lysophospholipids (LPs), particularly 10 specific LPs, in muscle flavor, while identifying pivotal genes regulating LP metabolism like *GLB1*, *PLD3*, *LPCAT2*, *DGKE*, *ACOT7*, and *CH25H* [80]. An analysis of the dorsal longitudinal muscle in the F1 generation, involving male Nanchu and Suffolk rams crossed with female Hu sheep, revealed 631 differentially expressed genes and 119 significantly altered metabolites. These influenced muscle development characteristics and diversified meat quality. Genes like *MYLK3*, *MYL10*, *FIGN*, *MYH8*, *MYOM3*, *LMCD1*, and *FLRT1* were linked to muscle growth, while *MYH8* and *MYL10* regulated both fatty acid levels and meat quality [81].

The correlation analysis of meat traits in sheep, evident in these studies, not only aids in identifying differences in sheep meat quality but also systematically and comprehensively dissects the underlying mechanisms and regulatory networks governing meat traits. This approach forms a scientific foundation for elevating and refining sheep meat quality, ultimately contributing to the advancement of sheep breeding practices.

### 3.2. Wool Traits

Hair, a defining feature among mammals, serves multifaceted roles in thermoregulation, protection, sensory perception, and social interaction [82,83]. Cashmere, specifically derived from secondary hair follicles, stands as a highly esteemed textile material of considerable economic value [84,85]. The quality and yield of sheep wool are intimately linked to the characteristics and structure of hair follicles [86,87,88]. Integrated multiomics analysis plays a pivotal role in examining wool traits in sheep, furnishing a comprehensive understanding of the mechanisms and genetic underpinnings associated with these traits.

Delving into the molecular intricacies governing hair follicle development holds the key to enhancing wool-related genetic traits in sheep. For instance, an integrated study employing RNA-seq and whole-genome bisulfite sequencing (WGBS) probed the relationship between hair follicle differentiation genes, transcription factor genes, and DNA methylation levels. This research illuminated that certain hair follicle differentiation genes were initially repressed by methylation during the induction phase but underwent demethylation and expression during the differentiation phase, indicating a significant role of DNA methylation in hair morphogenesis [89]. Similarly, utilizing a combination of RNA-seq and methylome datasets, researchers scrutinized four genotypes of Merino sheep skin across various stages of hair follicle development. Their study delineated differential expression profiles and identified key transcripts and transcription factors, such as *KLF4*, *LEF1*, *HOXC13*, *RBPJ*, *VDR*, *RARA*, and *STAT3*, contributing to hair follicle maturation [90].

In essence, the utilization of multiomics approaches in sheep research yields invaluable insights into the development and morphology of hair follicles. Deciphering the intricate interplay between hair follicle differentiation genes, transcription factors, and DNA methylation levels through this methodology enriches our comprehension of the molecular mechanisms steering hair follicle development. Such insights hold immense value in advancing and refining wool traits in sheep.

### 3.3. Reproductive Traits

Reproductive traits in sheep serve as pivotal determinants of profitability for livestock producers globally [91]. These traits, signifying flock productivity, encompass crucial factors like wool production, litter size, litter weight, and lamb count [92,93]. Grasping the molecular underpinnings of fertility is indispensable for advancing sheep reproduction. The intricate nature of this trait necessitates a more comprehensive approach than conventional molecular biology methods can offer.

Multiomics analysis stands as a potent and insightful avenue to delve into reproductive traits in sheep, exploring genetic foundations, molecular mechanisms, and targets for breeding selection.

In a study amalgamating GWAS and RNA-Seq association analysis, discernible gene expression disparities between Small-Tailed Han sheep and Sishui fur sheep unveiled potential regulators of fertility and prolificacy, notably genes linked to the *TGFβ* pathway and *NOTCH2* [94]. Investigating the genes and proteins associated with litter size in sheep, studies focused on the transcriptome and proteome analysis of ovarian specimens revealed critical regulators, including *HSD17B1* and *MSTRG.28645*, governing hormone secretion affecting sheep fecundity [95]. Exploring metabolomic and proteomic shifts in uterine flushes (UF) from pregnant sheep unveiled 16 proteins through proteomic analysis and identified 8510 molecular signatures using metabolomic analysis. These proteins and metabolites were identified as contributors to fetal nourishment [96]. Scrutinizing the postconception impact of diversely energized maternal diets on WGBS and RNA-Seq patterns in sheep offspring yielded valuable insights. This investigation unearthed correlations between gene expression and both inter-/intragenic methylated regions. Notably, the association of Intragenic Differentially Methylated Regions (DMRs) with the expression of neighboring genes unveiled the intricate interplay between DNA methylation and gene expression, offering crucial insights into the regulatory dynamics governing reproductive performance [97]. Employing a multiomics approach yielded insights into amniotic fluid transport pathways, revealing nine transport-related pathways and four groups of differentially expressed transcripts and proteins. This research illuminated the regulation of amniotic intramembranous transport and the role of transport mediators [98]. Investigating the connection between *FecB* locus polymorphisms and ovulation rate and litter size in sheep unveiled increased GnRH content during follicular development in specific ewes, correlating with more mature follicles [99]. Examining ovarian differences in Hu sheep with varying fertility and genotypes (*Fec^BB^* and *Fec^B+^*) unraveled significant disparities in DNA methylome and RNA-Seq between high-yielding and low-yielding individuals. Systematic integration analysis disclosed a negative correlation between DNA methylation around the transcriptional start site and gene expression levels, providing insights into the molecular control of reproduction [100].

To summarize, multiomics association analysis offers a comprehensive view of the intricate molecular regulatory networks governing sheep reproductive traits. This approach plays a pivotal role in refining reproductive management, optimizing performance, and fostering sustainable development within the sheep industry.

### 3.4. Ovine Physiology 

Joint multiomics analyses have far-reaching applications in sheep research, extending well beyond reproductive traits and encompassing various facets crucial to understanding sheep biology. This integrative approach holds immense value in deciphering mechanisms linked to adaptation to challenging environments and fortifying disease resistance in sheep. Integrating genomics, transcriptomics, and metabolomics data allows for the identification of pivotal genes and metabolic pathways associated with adaptive traits and immune responses [101,102]. This knowledge forms the bedrock for breeding sheep tailored to specific environments and possessing enhanced disease resistance.

Regarding atrial fibrillation (AF), a study conducted on cardiomyocytes (CM) isolated from sheep models with AF explored the impact of differing rapid intervals. Transcriptional–proteomic analyses unveiled differences in genes linked to mitochondrial augmentation, chromatin modifications in atrial CM, neural function, and cell proliferation during early AF stages [103]. These findings contribute significantly to our comprehension of AF’s molecular mechanisms. In a comprehensive analysis of rumen epithelial morphology, RNA-seq data, microbiology, and metabolomics in a Tibetan sheep model, adaptive changes in rumen epithelium during the cold season were revealed. Key insights surfaced, such as the significant upregulation of cytochrome P3 pathway epithelial gene-GSTM450 expression, associated with xenobiotic metabolism, and the downregulation of harmful metabolites. Additionally, the upregulation of the *TLR5* gene in the Legionnaires’ disease pathway and the downregulation of *CD14* gene expression were observed, elucidating the adaptive changes in response to cold environments [104]. Concentrating on E.coli-F17, a prevalent pathogen inducing diarrhea in livestock and poultry, a comprehensive multiomics approach involving microbiome, metabolome, and RNA-seq analyses was employed in lamb models. Significantly different metabolites between lambs resistant or susceptible to E.coli-F17 were identified, offering insights into lipid metabolism and potential biomarkers for infection [105].

In summary, multiomics analysis in sheep research plays a pivotal role in unraveling the complexities of sheep diseases and physiological adaptations. By scrutinizing changes in gene expression, protein composition, and metabolite levels, we can deepen our understanding of pathogenesis, paving the way for the development of novel strategies and targets for disease prevention, diagnosis, and treatment.

## 4. Conclusions

Extensive investigations into meat, wool, and reproductive traits, among other facets of sheep physiology (summarized in Figure 2), underscore the efficacy of multiomics association analysis in unraveling complex biological functions and physiological mechanisms in sheep. While this approach lays a robust scientific groundwork and provides a theoretical roadmap for advancing sheep research and industry practices, acknowledging prevailing limitations is critical. 

A significant challenge lies in the incomplete assembly of the sheep genome, notably the absence of a telomere-to-telomere (T2T) reference genome, which constrains the depth of trait analysis. Attaining a T2T reference genome is pivotal to enable more comprehensive future research. Furthermore, many current studies have predominantly focused on pairwise associations between two omics layers, lacking a holistic interpretation of interactions among multiple omics layers for specific traits. Addressing this gap through comprehensive analyses is imperative. Despite the complexities associated with multiomics joint analysis, such as data integration challenges and stringent sample quality requirements, this approach is indispensable for advancing genetic improvements, enhancing health management, and promoting sustainable sheep utilization. Technological advancements, especially in high-throughput sequencing, are crucial for the upcoming T2T genome assembly. Platforms like PacBio Revio, offering ultralong read lengths, and the reduced costs of sequencing technologies like those provided by Oxford Nanopore Technologies (ONT), play a pivotal role in this progress. Future directions encompass the application of cutting-edge techniques such as stem cell breeding, embryonic gene editing, and innovative gametes and embryo engineering. These advancements are poised to mitigate current limitations in multiomics association analysis and provide essential genetic insights for molecular breeding. Such progress will be pivotal in enhancing sheep genetics and breeding efforts, propelling the development of the sheep industry.

## Figures and Tables

**Figure 1 animals-14-00273-f001:**
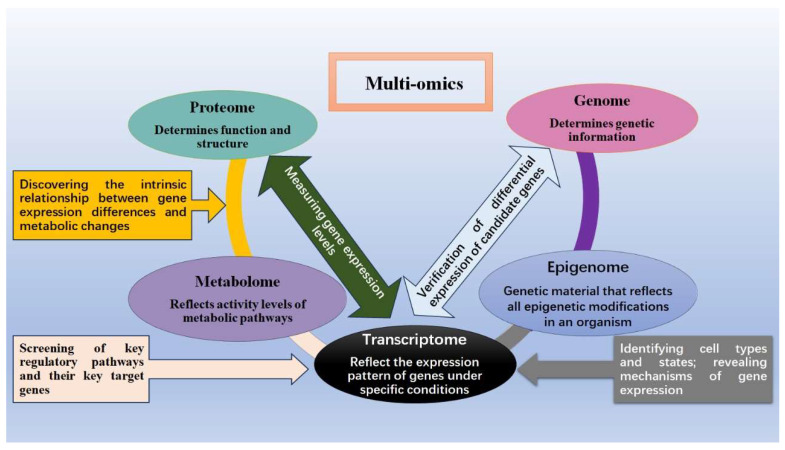
Connection diagram between multiple omics.

**Figure 2 animals-14-00273-f002:**
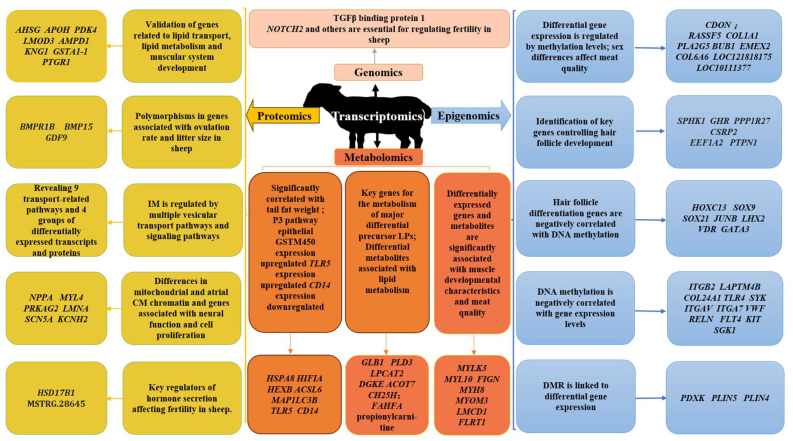
Schematic diagram summarizing the application of multiomics joint analysis in genetics and breeding of sheep.

**Table 1 animals-14-00273-t001:** Research progress on sheep genome assembly.

Breed	Genome Versions	Sequence Size/Gb	Contig N50	Scaffolds N50/Mb	Sequencing Technology	Time	Reference
Texel	Oar_v3.1	2.61	40 KB	100	Illumina	2014	[20]
Texel	Oar_v4.0	2.6	150.5 KB	100	Illumina; 454 PacBio	2015	GCA_000298735.2
Rambouillet	Oar_rambouillet_v1.0	2.9	2.6 Mb	107.7	HiSeq X× Ten PacBio	2017	GCA_002742125.1
Mouflon	Platanus	2.69	110.1 KB	10.4	Illumina	2020	[21]
Tibetan	CAU_O.aries_1.0	2.7	74.6 Mb	105.2	PacBio	2021	GCA_017524585.1
Rambouillet	ARS-UI_Ramb_v2.0	2.63	43.2 Mb	101.3	Illumina	2022	[22]
Dorper	Oar_v4.0; ARS-UI_Ramb_v2.0	2.64	73.33 Mb	--	Illumina PacBio	2022	[23]
Kazakh	ASM2243284v1	2.9	73.4 Mb	96.2	PromethION	2022	GCA_022432845.1
Dorset	ASM2241691v1	2.9	92.4 Mb	96.5	Ilumina	2022	GCA_022416915.1
Romanov x Dorper	Oar_ARS-UKY_WhiteDorper_v1.0	2.6	61.8 Mb	95.6	PacBio	2022	GCA_022244695.1

**Table 2 animals-14-00273-t002:** Application of Mono-Omics in Sheep Genetics and Breeding.

Genes/Metabolites	Omics Type	Mutant Site/Metabolic Pathway	Traits	Reference
*ASIP*	Genomics	g.100-105delAGGAA g.10-19delAGCCGCCTC g.5172T > A	Hair color	[24]
*RXFP2*	Genomics	3′UTR	Hornless	[25]
*CPOX* *KCNH1* *CPQ*	Genomics	g.178,730,623 T > G g.75,716,237 C > G g.88,323,841 A > G	Ribcage	[26]
*HOXB13*	Genomics	5′UTR	Tailed	[27]
*MAPT DLK1 DIAPH1 NR4A1*	Epigenomics	promoter region	Muscle growth metabolism	[38]
*HOXC9*	Transcriptomics	3′UTR	Caudal fat deposition	[49]
APOA2 GALK1 ADIPOQ NDUFS4	Proteomics	---	Caudal fat deposition	[59]
Amino acids, MMA Methylmalonic acid	Metabolomics	biosynthesis of amino acids; biosynthesis of unsaturated fatty acids	meat quality	[72]
Amino acids fatty acyl groups glycerophospholipids	Metabolomics	protein digestion and absorption; aminoacyl-trNA biosynthesis; carbon metabolism	Succulent	[73]

## Data Availability

No new data were created or analyzed in this study. Data sharing is not applicable to this article.

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
