# Peer review of "Pan-Omics in Sheep: Unveiling Genetic Landscapes"

_animals, 2024, doi:10.3390/ani14020273_

Round 1

Reviewer 1 Report

Comments and Suggestions for Authors

The manuscript of Li et al. under the title "Pan-Omics in Sheep: Unveiling Genetic Landscapes" is a very well written review that summarises comprehensively the multiomics achivements in sheep and how they can help to improve the breeding strategies. 

I have only minor suggestions: 1. please check if the colors of the figures are friendly for the color-blind people.

2. I use "genome-wide association study" for the first time in the line 132. So, there it should be accompanied by the abbreviation (GWAS). Then, later in the text use only the abbreviation GWAS. For example, in the line 378.

3. It may be not clear for the general reader not very familiar with sheep breeding, why you (and other studies) put so much emphasis on the tail fat. Why sheep tail is so important? Could be useful to explain a bit.

Comments on the Quality of English Language

English is good in general. Sometimes it is a bit repeatitive in adjectives ("pivotal", "unveiling", etc.)

Reviewer 2 Report

Comments and Suggestions for Authors

Dear authors,

I will recommend your manuscript for publication, but I has some questions for edition.

L9 - Simple Summary is larger then Abstract. All two these parts consist too much information about Omics technologies then its employing in sheep breeding.

L107 – I recommend add OAR 3.1 version in table

L133 – In genomic techniques add information about Illumina BeadChip research, because main candidate genes findings based on it.

In general – I recommend add part about Metabolic Pathway Analysis and about Milk Production in sheep.

Regards,

Reviewer 3 Report

Comments and Suggestions for Authors

Review on the manuscript titled “Pan-Omics in Sheep: Unveiling Genetic Landscapes” by Li et al., 2023

                The authors present a review on the Multi-omics (Pan-Omics) data state for the sheep breed with the aim of using the data for breeding. The omics they address in manuscript are: genomics, epigenomics, transcriptomics, proteomics, metabolomics, and microbiomics.

                As genomics segment state, they present the genome assemblies locations in nine sheep  breeds in Table 1.  Epigenetic variables include histone modifications, DNA methylation, and RNA modifications. They listed the genes shown to vary in epigenetic marks in Epigenetic chapter.

Transcriptomic studies section also outlines the number of genes/networks subject to impact the industrial traits of the species.

Next follows proteome methodology with the target genes elucidation in a range of previous projects [47-54], including diet tolerance and response. Metabolomics section also addresses review of several projects [55-66].

     Table 2 presented top projects on Genomics (4), epigenomics (1) transcriptomics (1) and metabolomics (1) while targeting certain traits (genetic traits: Hair color, Hornless, Ribcage, Tailed) epigenetic target genes (Muscle growth Metabolism), physiological ones (Caudal fat deposition, Succulent).

Next follow the chapters related to specific sheep traits (3. Research and Implementation of Multi-Omics Integration in Sheep Production), including Meat Traits (3.1), Wool traits (3.2), reproductive traits (3.3), and other physiological traits (3.4) where the authors depict corresponding studies.

Next follows database-like block scheme of the knowledge on the sheep in Fig. 2, with consequent remarks chapter 4 on challenges and future directions.

Overall, the authors compiled/referred the significant number of articles, and thus make the solid attempt in embracing the area of omics in sheep species. Also, they outlined valuable resources for sheep genome/traits analysis. Still there is a range of points for the manuscript listed below.

1)      I have a question on the scopes and objectives of the paper. While the authors discussed  the results in ‘omics’ area, Figure 2 comes afterwards as a major message hard to interpret, and in many ways not elaborated, this way rather unlinked with the previous text. The central object in Fig. 2. Diagram stands for RNA-seq as a major node. It connects with “proteomics” , “WGRS” (whole genome research studies??), and topped and bottomed with “metabolomics” nodes. We can see also GWAS node not represented in the previous text and hardly related to omics entity. Green, blue, and red boxes denote some statements  (results?) with genes sets (not related to Table 1) apparently supporting the statements. Microbiome is not represented in the previous text. The figure title declares it describes “multi-omics joint analysis in genetic and breeding of sheep”.

Looks like the authors got tired of reviewing studies literally, and rushed to produce the block-schema with no elaboration. It should be done explicitly, let it be in supplementary.

2)      There is a single referencing to Fig. 2 further on, manifesting that “…investigations into meat, wool, and reproductive traits, among other facets of sheep physiology (summarized in Figure 2) underscore the efficacy of multi-omics association analysis”, while the Fig. 2 caption mentions “genetics and breeding”. There is a long way between the physiological pathways and genetics/breeding. Physiology provides the target genes, but hardly many of them could be subject to selection due to the ages-long previous selection still based on genetically manifested phenotypes. From my experience, I saw most of the variable loci already highly selected to the beneficial allele frequency 95%.

3)      If the work targets breeders as declared in Fig 2 caption, it should stress a)  the genetic part of the ‘omics’, with most cost efficient approaches between various omics. The authors implicitly declared it RNA-seq (Fig. 2), but failed to explain it in terms of cost efficiency/relevancy compared to other omics. E.g., while ‘proteomics’ logically follows RNA-seq in terms of biology (as depicted in Fig. 2), it uses completely different approaches/methods which are more cost-consuming.

4)      I do believe that there is a high relevance creating/mentioning a knowledge base on sheep multi-omics /literature reviews (unless it already exists), where the user can access the information he’s interested in directly with a few clicks. The target Fig. 2 stands inappropriate in these terms due to an overload and poor structuring (it’s actually a multileveled task). So, it’d be much better to link the review to some user-friendly knowledge base on the sheep. It’d be much better providing such a tool, or just mentioning it in some form, since I feel the manuscript structure/content is ill conceived due to too multiple facets of the subject hard pooling together without basic segmentation, or hierarchical computer application/database.

5)      The authors should weight all ‘omics’ routines and evaluate the comparative costs, since it’s the pivotal issue for breeders. E.g. In proteomics section, the authors mentioned work [53]: “Comparative proteomic analysis of rumen epithelial tissues across different sheep ages revealed 4523 proteins, indicating the involvement of processes like glutathione, the Wnt signaling pathway, and the Notch signaling pathway in rumen epithelial cell growth [53]”. It is known that proteomics analysis is a cost consuming one compared to transcriptomics. The analysis cost mentioned in [53] would be order of magnitude less had they performed transcriptome analysis in this instance (for this result).

Based on the notes, I suggest first splitting the ‘omics’ analysis for breeders into two major logical subsections: a) elucidation of molecular targets for particular traits using gene network pathways assessment and b) analyse target genes on the genetic/epigenomic/rna-modification/GWAS variations. Practical points (omics most efficient strategies per costs) goes third. This way omics presentation would be more apprehensible to the breeders.

Comments on the Quality of English Language

Minor English corrections required.

Reviewer 4 Report

Comments and Suggestions for Authors

The present research provides interesting information regarding recent advancements in utilizing multi-omics technologies for sheep studies. A review of the content of the document has been carried out and 50% of the references are from the last 5 years, while 20% belong to the last year (2022). This greatly enriches this work. However, some changes need to be made before final publication.

Terms are repeated in the simple summary and summary delete in some of the two 9,10,11 and 24,25,26.

General comments: I recommend to go into the implications of the use of: “Genomics”, “Epigenomics”, “Transcriptomics”, “Proteomics” and “Metabolomics”.

In addition, I recommend making a "Table" of the main findings and most important works related to this area chronologically ("Genomics", "Epigenomics", "Transcriptomics", "Proteomics" and "Metabolomics") and how this tool has influenced the understanding of productivity and health of sheep. In "Table 2" they refer in part to this but it would be useful to augment it.

Some studies have been conducted on the construction of specific single nucleoside polymorphism (SNP) genetic marker sets for the identification of sheep breeds at the molecular level. It would be interesting to consider them in this review.

Yang J, Li WR, Lv FH, He SG, Tian SL, Peng WF, Sun YW, Zhao YX, Tu XL, Zhang M, et al: Whole-Genome Sequencing of Native Sheep Provides Insights into Rapid Adaptations to Extreme Environments. Mol Biol Evol 2016, 33, 2576-2592.

Hu XJ, Yang J, Xie XL, Lv FH, Cao YH, Li WR, Liu MJ, Wang YT, Li JQ, Liu YG, et al: The Genome Landscape of Tibetan  Sheep Reveals Adaptive Introgression from Argali and the History of Early Human Settlements on the Qinghai-Tibetan  Plateau. Mol Biol Evol 2019, 36, 283-303.

At the end of each paragraph of the topics "Genomics", "Epigenomics", "Transcriptomics", "Proteomics" and "Metabolomics", there is a small summary that I consider important. However, I consider that if the limitations of each of these tools could be included, it would be more enriching for the document. As it was done in "figure 2".

Specific comments:

1. Introduction

Line 59-68.- “Sheep represent one of the most  widespread livestock genetic resources globally, with over 1400 distinct breeds found  worldwide [9,10]. Their domestication traces back approximately 10,000 years, believed to have originated in the Middle East or Mesopotamia in Asia [11]. Known for their docile  nature, sheep offer valuable resources, providing wool for clothing and high-quality meat  with significant taste and nutritional value, thus serving as a crucial source of sustenance  and textiles for humans [12]. Due to the considerable genetic diversity among sheep, the  implementation of comprehensive multi-omics approaches in studying their genetic  information and associated functionalities across various levels plays a pivotal role in  improving their productivity and reproductive performance”. I recommend starting with this paragraph and then mentioning "Multi-omics joint analysis" on line 39.

Line 59 add space between “reliability.Sheep”

Line 75.- add the point in "exploration".

2. Main Omics Analysis Techniques

2.1. Genomics

Line 88-91.- “Noteworthy methodologies in current genomic  research encompass de novo sequencing, re-sequencing, and streamlined genome  sequencing. The advent of high-throughput sequencing technology has notably propelled  genomics, facilitating the efficient acquisition and analysis of vast genomic datasets”. I recommend supporting this sentence with a citation (author and year).

2.3. Transcriptomics

Line 178.- analyses have been carried out to determine "the hypothalamus regulates ovulation under the effect of the FecB mutation", it would be interesting to consider it.

Chen S, Guo X, He X, Di R, Zhang X, Zhang J, Wang X, Chu M. Transcriptome Analysis Reveals Differentially Expressed Genes and Long Non-coding RNAs Associated With Fecundity in Sheep Hypothalamus With Different FecB Genotypes. Front Cell Dev Biol. 2021 May 20;9:633747. doi: 10.3389/fcell.2021.633747. PMID: 34095109; PMCID: PMC8172604.6

3. Research and Implementation of Multi-Omics Integration in Sheep Production

General comments: it would be convenient to mention a brief context of these "Multi-Omics" studies before talking specifically about each area of use.

Specific comments: A diagram summarizing the applications of multi-omics joint analysis in sheep is shown in Figure 2. It is well done, however I suggest referencing each of the genes, metabolites, etc., indicated in the diagram.

Round 2

Reviewer 3 Report

Comments and Suggestions for Authors

I have no further comments on the manuscript